

# GREEN: A new Global Radiation Earth ENvironment model

Angélica Sicard[1], Daniel Boscher[1], Sébastien Bourdarie[1], Didier Lazaro[1], Denis Standarovski[2], Robert Ecoffet[2]

[1] ONERA, The French Aerospace Lab, Toulouse, France
[2] CNES, The French Space Agency, Toulouse, France

*Correspondence to*: Angélica Sicard (angelica.sicard@onera.fr)

**Abstract.** GREEN (Global Radiation Earth ENvironment) is a new model providing fluxes at any location between L*=1 and L*=8 all along the magnetic field lines and for any energy between 1 keV to 10 MeV for electrons and between 1 keV and 800 MeV for protons. This model is composed of global models: AE8/AP8 and SPM for low energies and local models:
SLOT model, OZONE, IGE-2006 for electrons and OPAL and geostationary model for protons. GREEN is not just a collection of various models, it calculates the electron and proton fluxes from the more relevant existing model for a given energy and location. Moreover, some existing models can be updated or corrected in GREEN. For examples, a new version of the SLOT model is presented here and has been integrated in GREEN. Moreover, a new model of proton flux at geostationary orbit (IGP), developed few years ago is also detailed here and integrated in GREEN. Finally a correction of
AE8 model at high energy for L*<2.5 has also been implemented.

## 1 Introduction

The well known AP8 and AE8 NASA models [Vette, 1991; Sawyer and Vette, 1976] are commonly used in the industry to specify the radiation belt environment. Unfortunately, there are some limitations in the use of these models, first due to the covered energy range, but also because in some regions of space, there are discrepancies between the predicted average

values and the measurements. Moreover, new US models AE9/AP9 were developed a few years ago [Ginet et al., 2013]. These models are better than AE8/AP8 in some cases but are still very controversial in some regions of radiation belts. Therefore, our aim is to develop a radiation belt model, covering a large region of space and energy, from LEO altitudes to GEO and above, and from plasma to relativistic particles. The aim for the first version of this new model is to correct the AP8 and AE8 models where they are deficient or not defined. Ten years ago we developed the IGE-2006 model for

geostationary orbit electrons [Sicard-Piet et al., 2008]. This model was proven to be more accurate than AE8, and used commonly in the industry, covering a broad energy range, from 1keV to 5MeV. From then, a proton model for geostationary orbit, called IGP, was also developed for material applications and is presented in this paper. These models at geostationary orbit were followed by the OZONE model [Bourdarie et al., 2009] covering a narrower energy range but the whole outer electron belt, a SLOT model [Sicard-Piet et al., 2014] to asses average electron values for 2<L*<4, and finally the OPAL

model [Boscher et al., 2014], which provides high energy proton flux values at low altitudes. As most of these models were



developed using more than a solar cycle of measurements, these ones being checked, cross calibrated and filtered, we have no doubt that the obtained averages are more accurate than AP8 and AE8 for these particular locations. These local models were validated along different orbit with independent data sets or effect measurements.

In order to develop a new global model called GREEN, with GREEN-e for electrons and GREEN-p for protons, we will use
a cache file system to switch between models, in order to obtain the most reliable value at each location in space and each energy point. Of course, the way the model is developed is well suited to future enhancement with new models developed locally or under international partnerships. The first beta version of the GREEN model is presented in this paper.

## 2 Development of the model

### 2.1 Main principles

GREEN is a new model composed of different global and local models. The first step of the development was to define a list of the more relevant models in the case of electrons and an other one for protons. These two lists can be expanded and modified at any time. GREEN-e is composed of AE8 [Vette, 1991], SLOT model [Sicard-Piet et al., 2014], OZONE [Bourdarie et al., 2009], IGE-2006 [Sicard-Piet et al., 2008] and SPM for the lower energies [Ginet et al., 2013]. GREEN-p is composed of AP8 [Sawyer and Vette, 1976], OPAL [Boscher et al., 2014] and SPM [Ginet et al., 2013]. The second step
was to define a 3-dimensional grid in energy (Ec), $B_{local}/B_{eq}$ (with $B_{local}$ the local magnetic field and $B_{eq}$ the equatorial magnetic field) and L*. This grid represents the global architecture of GREEN. This 3D grid (Ec, $B/B_{eq}$, L) is the same as the one used for the physical model Salammbô [Herrera et al., 2016, and reference there in] with 133 steps in L* (between L*=1 and L*=8), 133 steps in $B/B_{eq}$ and 49 steps in energy and has not been chosen randomly. After verification, this grid allows to reproduce as best as possible the results of the most binding model (as OPAL for example). Obviously the energy grid is
different for GREEN-e and GREEN-p. Then, fluxes from each model integrated in GREEN have been calculated on this 3D grid. Taking into account that some local models that composed GREEN give only flux integrated in energy, only this kind of flux are provided by GREEN [$cm^{-2}.s^{-1}$]. Finally, a priority order of the different models has been established according to space location and energy to provide the most reliable value of flux. The last step is to calculate flux for a given energy and a given location by interpolating in the 3D grid of the most reliable model.




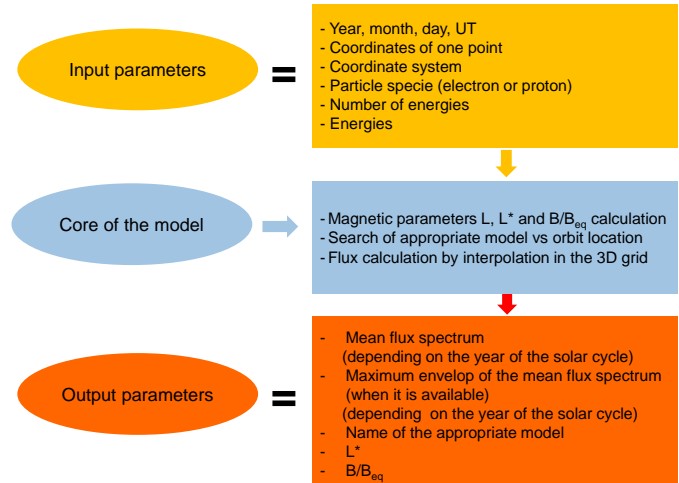

**Figure 1: Input and output parameters of the GREEN model**

Figure 1 is a scheme describing all the input parameters, the core of the model and all the output parameters of GREEN. One

5    of the features of GREEN is that it provides fluxes depending on the year of the solar cycle and not just two states as in the case of AE8 (AE8 MIN and AE8 MAX). Moreover, when it is possible, GREEN provides also the maximum envelop of the mean flux, depending also on the year of the solar cycle, due to the variation from one solar cycle to another (as explained in details for IGE-2006 [Sicard-Piet et al., 2008]).

**2.2 GREEN-e**

10   In this section, the electron part of GREEN, GREEN-e, is described in details. Figure 2 represents energy and L coverage of the different models integrated in GREEN-e. It is important to keep in mind that most of the models are defined in terms of L* calculated with IGRF+Olson Pfitzer magnetic fields models except AE8. Indeed, when AE8 is used, the L parameter must be calculated with Jensen and Cain magnetic field model [Vette, 1991].

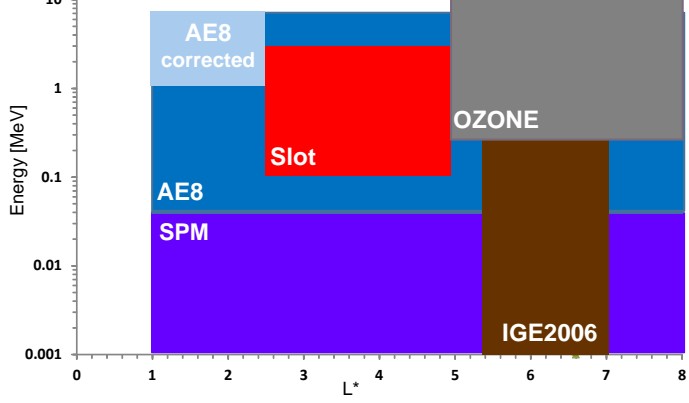



**Figure 2: Energy and L coverage of the different models integrated in GREEN-e**

**2.2.1 AE8 and SPM**

As it is mentioned on Figure 2, AE8 and SPM are used by default. This is the case for SPM model at low energy (<30 keV) except at geostationary orbit when IGE-2006 is preferred and for AE8 at higher energy (>30 keV) outside the coverage of

5      the SLOT model, OZONE and IGE-2006. SPM is a model with no solar cycle dependence, thus electron fluxes resulting from this model are considered constant along the solar cycle. For AE8, two versions exist: AE8 MAX for the solar maximum and AE8 MIN for the solar minimum. It is common to consider a full solar cycle of eleven years with 4 years of solar minimum (2 years before the minimum and 2 years after) and 7 years of solar maximum. Thus, in GREEN, when AE8 is the preferred model, the appropriate version of AE8 (MIN or MAX) is taken according to the year chosen by the user.

As it is mentioned on this figure, for L< 2.5 and energy greater than 1 MeV, we choose to use a corrected version of AE8. Indeed, in a previous study, Boscher at al. [2017] showed that high energy electron fluxes (> 1MeV) predicted by AE8 are overestimated in the region for L*<2.5. It is difficult to estimate the error made by AE8 but this study aims at showing that in this region and for high energy, the physical model Salammbo provides electron fluxes in agreement with in-situ

15     measurements. Thus, in this version of GREEN-e model, AE8 fluxes have been corrected, that is to say divided by a given factor, calculated using the Salammbô model. The Salammbo model is not perfect everywhere but it has been proved that the decrease of electron flux with energy is good [Boscher et al., 2017]. Thus, when electron fluxes from AE8 are higher than those provided by Salammbô, they are divided by the ratio between the both, up to a factor 100, in order to limit the correction (Figure 3). This correction is not perfect but allows to better estimate high energy electron flux in the region

20     L*<2.5.

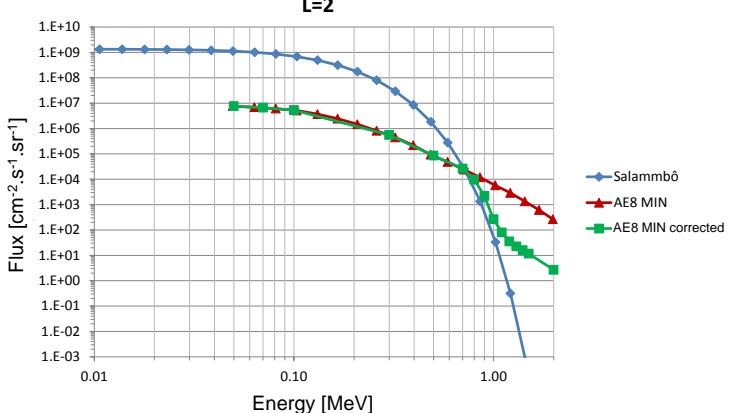

**Figure 3: Electron fluxes provided by Salammbô in blue (averaged on years during solar minimum), by AE8 MIN in red and AE8 MIN corrected in green at L=2**





### 2.2.2 SLOT model

Figure 2 shows also that the SLOT model is available from L*=2.5 to L*=5 and for energies from 100 keV to 3 MeV. The SLOT model developed in 2013 [Sicard-Piet et al., 2014] was a mean model. This first version has been updated in 2017 and is described here. As explained in a previous paper [Sicard-Piet et al., 2014], the SLOT model is based on the correlation

between the flux dynamics in LEO orbit with NOAA-POES data and the flux all along the magnetic field line. The first change in the model is its spatial extension: the upper spatial limit of the SLOT model is now L*=5 against L*=4 before. Then, taking into account new measurements as those from Van Allen Probe (MAGEIS), correlation factors all along the magnetic field line have been recalculated, between L*=2.5 and L*=5. An example of correlation between NOAA-POES data and Van Allen Probe-A measurements is plotted on Figure 4 for electrons with energy >0.3 MeV and for L* between

3.7 and 3.8. This kind of correlation is made with all data used in the model and listed in [Sicard-Piet et al., 2014] plus Van Allen Probe. As explained by Sicard-Piet et al. [2014], these correlation factors are multiplied to the NOAA-POES data in order to obtain mean electron fluxes between >0.1 MeV and >3 MeV along all magnetic field lines between L*=2.5 and L*=5 (Figure 5).

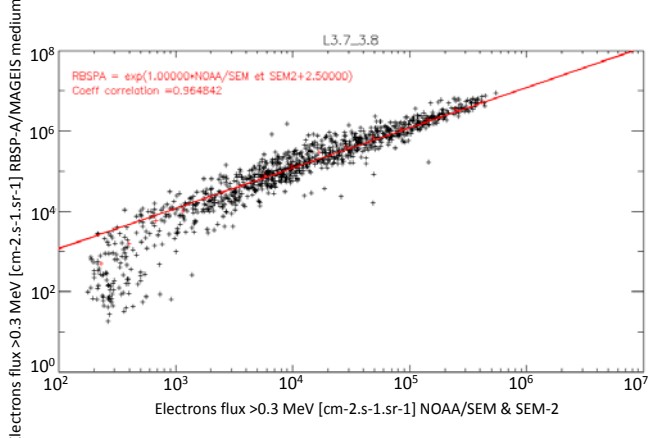

**Figure 4: Example of correlation between NOAA-POES and Van Allen probe-A flux for electrons >0.3 MeV and for L* between 3.7 and 3.8**



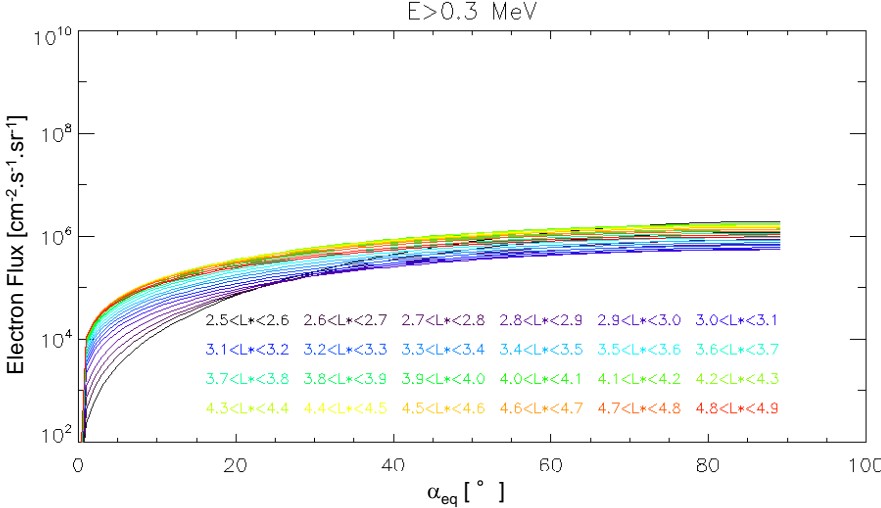

**Figure 5: Electrons flux along magnetic field lines (90° corresponds to equator) for energy > 0.3 MeV and for all L* intervals of the SLOT model (in color)**

Then, the most significant change in this new version of the SLOT model is the dependence of fluxes with the solar cycle. In order to have dependence with the solar cycle in GREEN-e, we have study in detail the dynamics of the measurements from all NOAA-POES spacecraft. As an example of this solar cycle dependence, Figure 6 representing >300 keV electron fluxes versus time from all NOAA-POES data is plotted. This figure shows clearly a correlation between the dynamics of NOAA-POES electron fluxes and the solar cycle (F10.7). The dynamics has been studied for four energy channels: >0.1 MeV, > 0.3 MeV, >1 MeV and > 3 MeV.

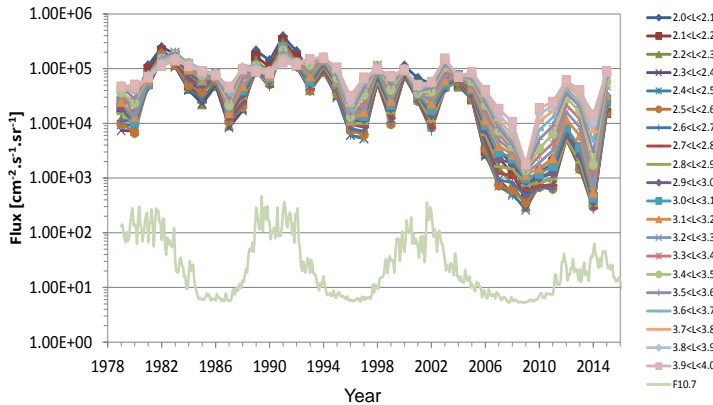

**Figure 6: Electron >300 keV fluxes versus time (1978-2015) from all NOAA-POES data for each L* intervals defined in the SLOT model. F10.7 is also represented in green**

Then, the flux dynamics along time between 1978 and 2015 has been represented on the eleven years on a solar cycle, from year -6 to year 4 with year 0 the solar minimum. The fluxes versus year of solar cycle at NOAA-POES orbit for electrons > 1 MeV and for each L* intervals is plotted on Figure 7. This modulation with solar cycle has been defined at LEO orbit for the



four energy channels of the SLOT model and has been applied to the mean flux all along the magnetic field lines, from low altitude to equator.

Finally, in order to take into account the modulation of flux from one solar cycle to another, this new version of the SLOT model provide mean flux for a given year of the solar cycle but also the maximum flux of the 3 solar cycles used in the model for this given year.

Thus, the new version of the SLOT model provide electron fluxes from 0.1 MeV to 3 MeV for all altitudes with L* between 2.5 and 5 with a dependence with the solar cycle.

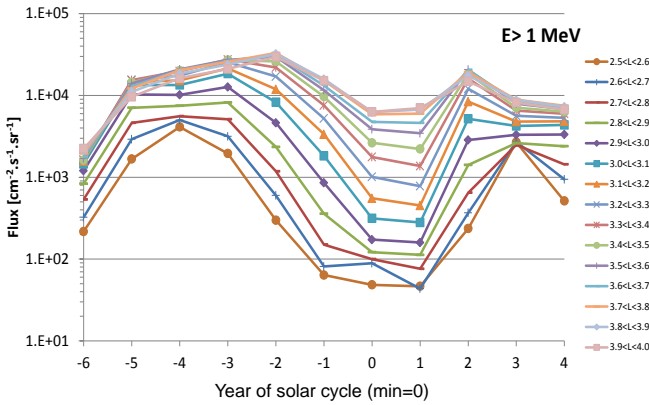

**Figure 7: Electron >1 MeV fluxes versus year of solar cycle from all NOAA-POES data for each L\* intervals defined in the SLOT model**

### 2.2.3 OZONE model

OZONE is valid for L*>4 and for energies greater than 300 keV. At geostationary orbit, OZONE agrees with IGE-2006 results, consequently OZONE will be used for energies > 300 keV at GEO orbit. The version of OZONE developed in 2009 [Bourdarie et al., 2009] was already depending on the solar cycle so no modification has been done on the model before the integration in GREEN-e.

### 2.2.4 IGE-2006 model

IGE-2006 is a specification model developed exclusively for geostationary orbit [Sicard-Piet et al., 2008]. This orbit is at a fixed altitude but is represented by a large L* range, between 5.7 and 7.1. As explained in Sicard-Piet et al. [2008], fluxes provided by IGE-2006 come from averaged fluxes measured by all available LANL spacecraft. In this version of GREEN-e, fluxes will be considered as a constant in this L* range. IGE-2006 is solar cycle dependent so no modification has to be done on the model before the integration in GREEN-e.





## 2.3 GREEN-p

In this section, the proton part of GREEN, GREEN-p, is described in detail. Figure 8 represents energy and L coverage of the different models integrated in GREEN-p. It is important to keep in mind that most of the models are defined in terms of L* calculated with IGRF+Olson-Pfitzer [Olson and Pfitzer, 1977] magnetic fields models except AP8. Indeed, when AP8 is

used, the L parameter must be calculated with Jensen and Cain magnetic field model for AP8 MIN and GSFC model for AP8 MAX [Sawyer and Vette, 1976].

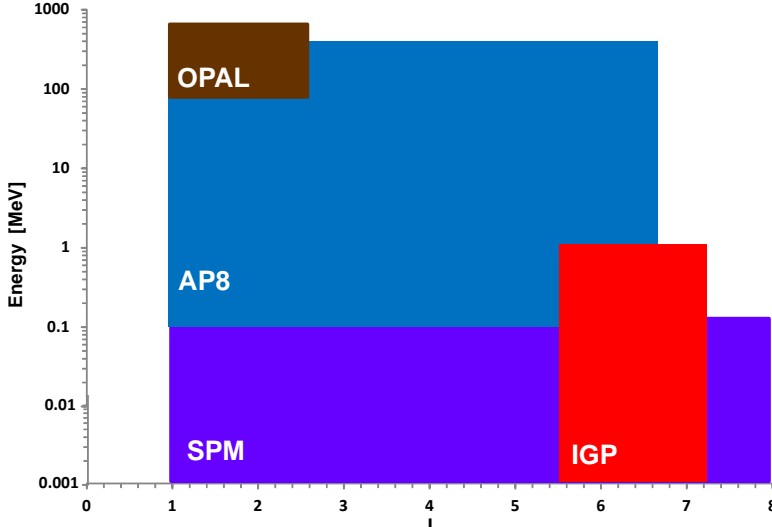

**Figure 8: Energy and L coverage of the different models integrated in GREEN-p**

**2.3.1 OPAL**

The first version of OPAL was a model valid for protons > 80 MeV and for altitude lower than 800 km, depending on the solar cycle [Boscher et al., 2014]. This year, a new version of OPAL has been developed at ONERA, using ICARE-NG measurements on board JASON-2 and JASON-3 [Boscher et al., 2011]. Now OPAL-v2 provides protons fluxes for energy between 80 MeV and 800 MeV up to the orbit of JASON spacecraft (1336 km). It is important to keep in mind that input

parameters of OPAL are the radio flux F10.7 of the Sun and the magnetic field of the given year. Figure 9 represents an example of protons flux spectrum $(cm^{-2}.s^{-1}.sr^{-1})$ at L*=1.3 near the magnetic equator ($\alpha_{eq}$=85.125°) resulting from OPAL-V2 (in blue) and AP8 MIN (in green). An extrapolation of OPAL-V2 at low energy is also represented in dashed line. We can observe that for this L* value, fluxes from OPAL-V2 are slightly higher than those from AP8 MIN while the extrapolation of OPAL-V2 is lower. However, this extrapolation has to be used with caution. This new version of OPAL (without

extrapolation) has been integrated in GREEN-p.



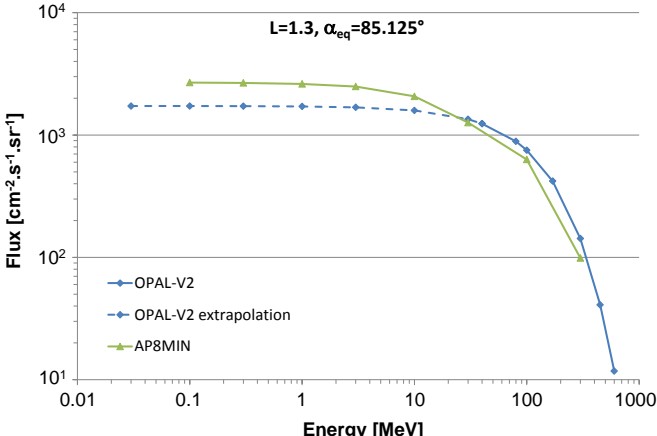

**Figure 9: Example of protons flux spectrum [cm$^{-2}$.s$^{-1}$.sr$^{-1}$] resulting from OPAL-V2 (in blue) and AP8 MIN (in green). An extrapolation of OPAL-V2 at low energy is also represented in dashed line**

### 2.3.1 IGP

Onboard the Los Alamos National Laboratory satellites, from July 1976 (launch of the satellite 1976-059) to June 1995 (end of the measurements on board 1984-129 and 1987-097), there was a detector named CPA (for Charged Particle Analyser), which covered the energy range 80keV-300MeV [Higbie et al., 1978 ; Baker et al., 1979]. To cover a larger energy range, we used also the measurements of the MPA (Magnetospheric Plasma Analyzer) detector on board LANL satellites being launched between September 1989 (launch of the satellite 1989-046) and November 1995 [McComas et al., 1993]. These

measurements cover roughly the energy range 0.1keV-38keV.

MPA measurements:

MPA measurements are globally of good quality. The temporal resolution is most of the time 86s, but it can be doubled (172s) for short periods of time. The detector aged with time; it drifted. This drift is compensated along time, but after several years it is impossible to measure the highest energies any more (typically after 10 years, it is impossible to obtain

measurements above 10keV). For the development of a proton specification model, data between 1keV and 32keV have been used. Fluxes below 1keV have not been used, due to uncertainties in the spacecraft potential determination. Thus, we determined monthly averages of the proton flux for each satellite. These monthly averages were made in order to analyse possible solar cycle or seasonal effects (linked to the magnetic field or to its activity). An example is given for the 1 keV protons in Figure 10. Some points as high as 2.3 10$^9$ MeV$^{-1}$cm$^{-2}$s$^{-1}$sr$^{-1}$ observed in June 1991 can be either due to the effect of

magnetic activity, a particular contamination during that period, or a (or several) bad point(s). Apart from these, no seasonal effect is observed in the flux curve, and if there is a solar cycle effect, it is very low. As the flux does not vary with time, an average spectrum was deduced from all the measurements, taking into account the number of points for each satellite.





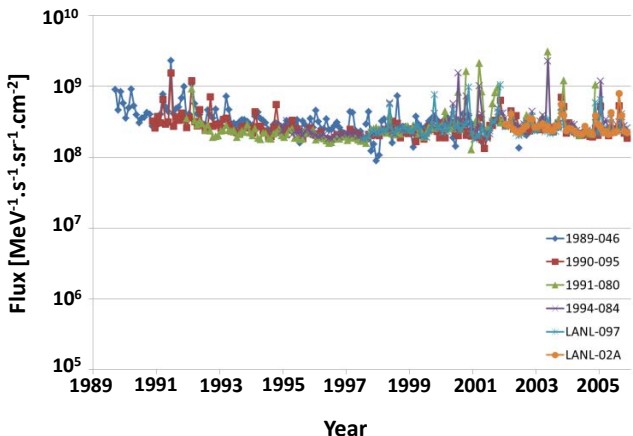

**Figure 10: Monthly average 1keV proton flux measured at GEO by the detector MPA on board the different LANL satellites**

CPA measurements:

The CPA instrument is in fact made of 2 different instruments: CPA-LoP and CPA-HiP which respond respectively to

5    protons in the range 73-512keV and 400keV-300MeV. The measurements are also globally of high quality. The time

resolution of the instrument is 10s, which means that the number of points is much higher. A monthly average for each

channel was produced. An example of this average is plotted on Figure 11 for 80 keV protons for each available LANL

spacecraft. From that figure, it appears that there is no seasonal variation in the 80 keV proton flux, and if there is a solar

cycle one, it should be small in the range covered by CPA-LoP (less than a factor of 2). We must note in this figure a few

10   low flux values which lie the general tendency of the curve; it is suspected that they are due to gain switches for that

particular channel and that satellite. We have not removed them, as the total average is not affected by these points.

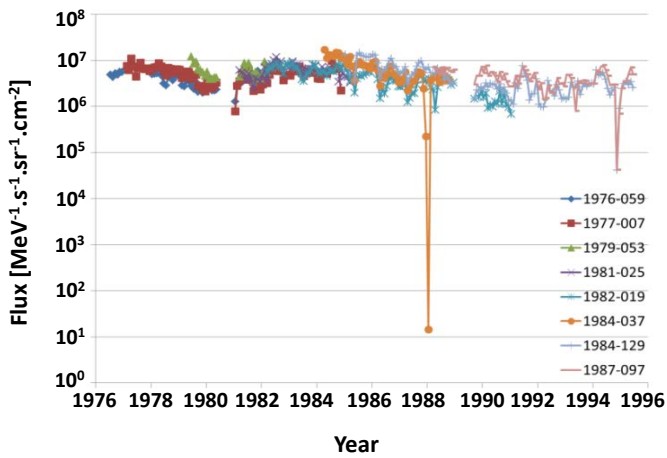

**Figure 11: Monthly average 80keV proton flux measured at GEO by the detector CPA-LoP on board the different LANL satellites**





As for MPA, a global average of all the points was performed, in order to obtain a global spectrum of protons from 1keV up to 1MeV at geostationary orbit. Above 1 MeV, data were not used because of the contamination by protons from solar flares.

IGP model:

Combining the part of the spectrum from MPA and CPA up to around 1 MeV leads to the Figure 12. The average fluxes are plotted, together with an error bar which corresponds to the maximum and minimum values obtained in the monthly averages from full time period and all spacecraft. Though a gap exists between the 2 instruments, it appears that both parts of the spectrum are consistent: at low energy the spectrum is very flat; it falls very quickly for energies greater than 50keV. We also compared in this figure the obtained spectrum with AP8 (for longitude 0°, AP8 MAX and MIN being equal in this

region) [Sawyer and Vette, 1976]. For unidirectional flux comparison, we divided AP8 flux by $4\pi$, the environment being nearly isotropic at geosynchronous orbit for trapped particles. We can see that the obtained spectrum is nearly consistent with AP8. In fact, near 1MeV, the main problem is to distinguish trapped particles from untrapped ones (solar protons and cosmic rays). That maybe explains part of the difference. Globally, while the obtained spectrum is nearly a power law, AP8 is more an exponential law, with a characteristic energy around 100keV.

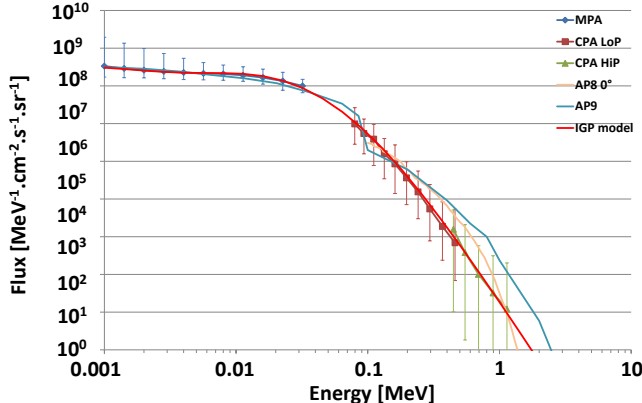

**Figure 12: Total spectrum of trapped protons deduced from MPA and CPA measurements on board different LANL satellites and from the model. Fluxes from AP8 and AP9 are provided for comparison.**

We tried to determine an empirical formula with all the average flux values. For the high energy part, we used a kappa function with 9keV characteristic energy and $\varkappa = 5.45$, not far from what was obtained by Christon et al. [1991] in the

plasma sheet. An exponential part (with 2keV characteristic energy) was added at low energy to fit the total spectrum:

$$flux = 4.10^8.\exp(-E/0.002) + 7.10^{10}.E.\left(1 + \frac{E}{5.45x0.009}\right)^{-6.45},$$

where E is the energy in MeV and the flux in MeV$^{-1}$cm$^{-2}$s$^{-1}$sr$^{-1}$.



The model result is compared to the average spectrum in Figure 12. This spectrum is very useful for deducing surface material degradation for satellites at geostationary orbit. It also can be used for dynamic physical model of the radiation belt proton to set a boundary condition.

This model is compared to the AP9-SPM NASA one [Ginet et al., 2013; Roth et al., 2014] also in Figure 12. The NASA
AP8 model was limited to energies greater than 100keV. As for AP9-SPM, it is at geostationary orbit clearly made by adding a high energy component, not so different than AP8, and a low energy component from SPM. At geostationary orbit, the low energy part comes from the same measurements we used: the MPA detector on board the LANL spacecraft and the two models are very close (the difference can be due to the interpolation used between channels). In AP9-SPM, the obtained spectrum is extrapolated to around 100keV, but it is possible that our way to connect the 2 parts of the model has to be
improved. With AP9-SPM, the two parts of the spectrum do not match together. Obviously, there is a discontinuity at around 100keV. Higher in energy, the spectra from AP8 and AP9 are not too different, up to around 500keV. Above this value, AP9 exceeds AP8 by a growing factor. The main problem for such energies is to distinguish in the measurements trapped and non trapped particles. We know from magnetospheric shielding calculations that for this energy range, both particles can be observed depending on the viewing direction. Looking to the East, trapped particles from the radiation belts
are observed while looking to the West, only cosmic rays and solar protons coming from outside the magnetosphere are observed. That is why in our analysis, no points were extracted for E>1.14MeV. The model just gives an extrapolation (reasonable as a power law).

## 3 Results and validation

Once each of the local models has been integrated into GREEN, we are able to calculate fluxes at any location between L*=1
and L*=8 all along the magnetic field lines and for any energy between 1 keV to 10 MeV for electrons and between 1 keV and 800 MeV for protons. Figure 13 and Figure 14 give an example of electron fluxes provided by the GREEN-e model in 1996 (solar minimum) and in 2003 (solar maximum) respectively, versus L* and energy at the equator. The different models used are also mentioned on the plot. These figures show clearly the influence of the solar cycle on the electron flux, particularly in the Slot region where fluxes are higher during solar maximum. Moreover, we can note that discontinuities
exist at the interface of the different models and have to be removed or at least smoothed in the future versions of GREEN-e.





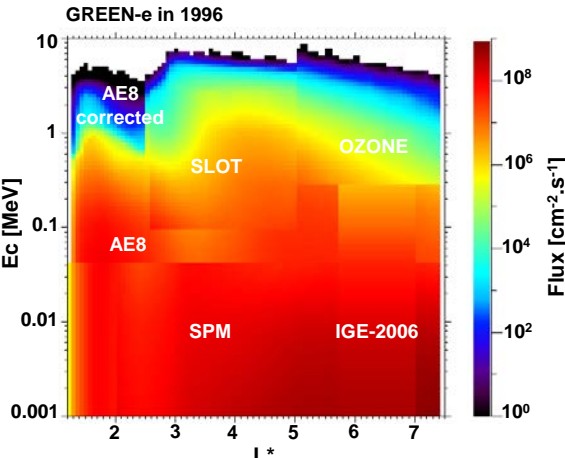

**Figure 13: Electron fluxes versus L\* and energy in 1996 provided by the GREEN-e model**

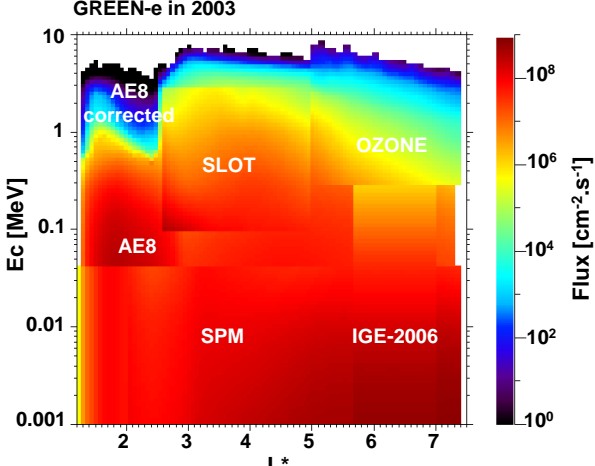

**Figure 14: Electron fluxes versus L\* and energy in 2003 provided by the GREEN-e model**

5  Now that the results of GREEN-e have been presented it is important to validate them. The first validation is done at MEO orbit by comparing electrons fluxes providing by GREEN-e and by MEO-V2 model [Sicard-Piet et al., 2006]. MEO-v2 model is not used in GREEN-e and is a good way to validate it. Figure 15 represents electron spectrum from GREEN-e in blue and from MEO-V2 model in red for the mean flux and in green for the upper envelop, for a whole solar cycle. This figure shows that electrons fluxes provided by the GREEN-e model are coherent with those resulting from MEO-V2 model:

10  equal or slightly higher than mean MEO-V2 fluxes and lower than upper envelop fluxes.

In order to validate fluxes at other orbits, a comparison between GREEN-e results and NOAA-POES measurements is done at LEO orbit. Figure 16 represents the mean electrons fluxes between 1999 and 2010 from NOAA-POES measurements (in dashed lines) for several energy channels (>30 keV, >100 keV, >300 keV, >1 MeV and > 3 MeV). Fluxes from GREEN-e,



calculated on a full solar cycle, are also plotted on the figure for comparison. This figure shows that beyond L*=2.5 fluxes resulting from GREEN-e are in agreement with NOAA-POES data, with less than a factor 3 between the both, particularly in the L-range of the SLOT model (2.5<L*<5) which is based of these data. At high energy (> 3 MeV) for L* >6, POES data seems to reach the background of the instrument, probably due to cosmic particles measurements, while fluxes from

GREEN-e model continue to decline while L* increase. We can note that for low energy (~30 keV), there is a big difference between GREEN-e and NOAA-POES measurements and that for some L* values this flux is lower than >100 keV flux, which is not usual. It is important to keep in mind that for low energy (~30 keV), electrons fluxes in GREEN-e come from AE8 while fluxes for higher energies come from the SLOT model and OZONE. This energy channel (~30 keV) would be a track of improvement of GREEN. Moreover, fluxes below L*=2.5 are not plotted in the figure because it is well known that

NOAA-POES data are contaminated by very high energy protons at low L* values [Evans and Greer, 2000].

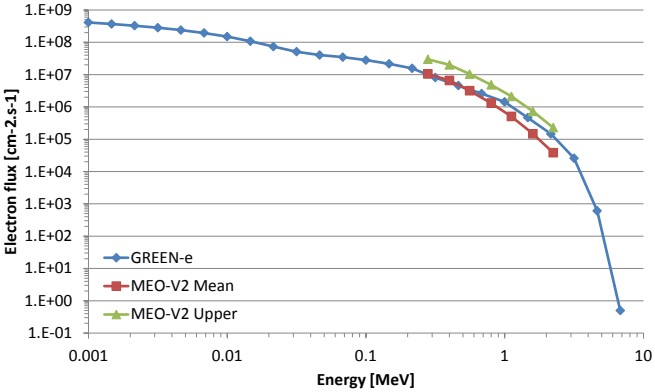

**Figure 15: Electron spectrum from GREEN-e (in blue) on a whole solar cycle compared to the mean (in red) and upper (in green) flux provided by MEO-V2 model**

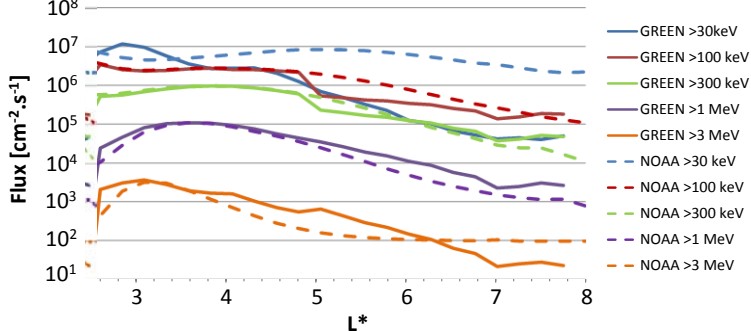

**Figure 16: Mean electrons fluxes at LEO orbit between 1999 and 2010 from NOAA-POES measurements (in dashed lines) and from one solar cycle in GREEN-e (in full lines) for several energy channels**

The same kind of comparison have been made between GREEN-e results and JASON-2 data for E>2.02 MeV electrons for L*< 2.5 between 2009 and 2015 and is plotted on Figure 17. 2009 to 2015 corresponds to years 0, 1, 2, 3, 4, -5 and -6 of the solar cycle (0 is the year of the minimum). On Figure 17, fluxes are an averaged of results from GREEN-e for these years of



the solar cycle. This graph shows first that there is a discontinuity in GREEN-e model at L*=5, at the interface between the SLOT model and OZONE. Some efforts will be made to remove this kind of discontinuity in the next version of GREEN. However, what we want to highlight with this plot is the difference between GREEN-e results and JASON-2 measurements for low L* values (L*<3.5). Electron flux measured by JASON-2 at this energy are much lower than the one provided by

GREEN in this region while Figure 16 showed a very good correlation between GREEN and NOAA measurements in the same region (L*<3.5). Why was there an agreement between the results of GREEN and NOAA that no longer appears with the JASON-2 measurements? Is this due to the difference of altitude between the two spacecraft (800km for NOAA and 1336 km for JASON-2)?

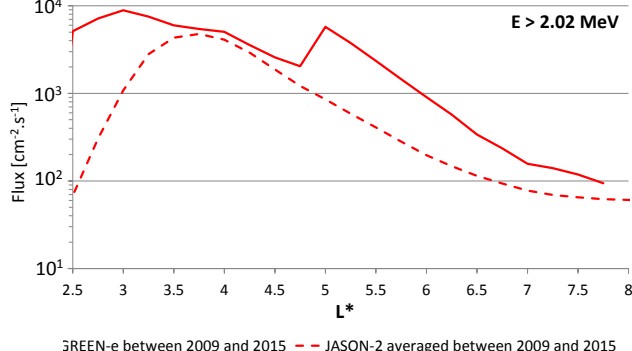

**Figure 17: Mean electrons fluxes at JASON-2 orbit between 2009 and 2015 from GREEN-e (in full lines) and JASON-2 measurements (in dashed lines) for E>2.02 MeV electrons.**

In order to illustrate the reason of this difference between JASON-2 measurements and GREEN results, Figure 18 has been plotted. It is the same figure than Figure 16 but not during the same period of time: 2009 to 2015 for Figure 18 against 1999 to 2010 for Figure 16. This figure shows that the comparison between in-situ measurements and GREEN results depends on

the period of time. If the period of time of in-situ measurements is long enough (several solar cycle) or is representative of a mean flux, data will easily be compared to GREEN- results. On the other hand, if the period of time of in-situ measurements is too short compared to a solar cycle, or is during a very quiet solar cycle, which is the case for JASON-2 measurements, comparison with GREEN flux will not be so easy. So, the difference of flux at L*<3.5 between GREEN-e and JASON-2 data on Figure 17 or between GREEN-e and NOAA-POES data on Figure 18 is clearly due to the period of time, which

correspond to very quiet years, not representative of a mean solar cycle.


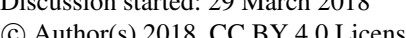


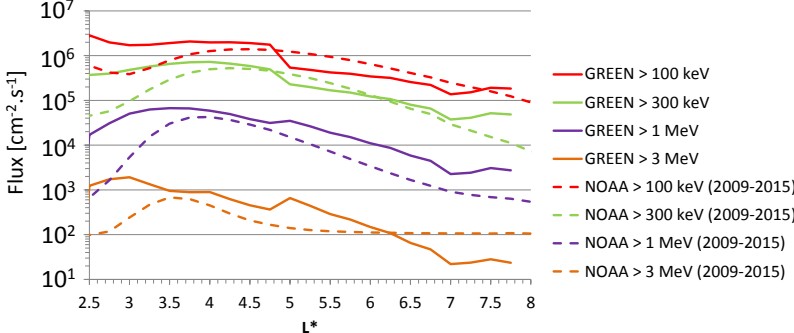

**Figure 18: Mean electrons fluxes at LEO orbit between 2009 and 2015 from NOAA-POES measurements (in dashed lines) and from GREEN-e (in full lines) for several energy channels**

Concerning the model GREEN-p, it is much less finalized than the electron version GREEN-e because only OPAL model, which has a narrow spatial coverage, has been implemented in addition to AP8 and SPM. It is really difficult to measure protons of energy around MeV in the radiation belts because of the predominant presence of the electrons which very often contaminate the data. Thus, by lack of good quality data in sufficient number it is difficult to develop a model of protons for

10 energies around MeV. Some efforts will be made in the near future to improve the modelling of MeV protons in GREEN-p and compare the results with measurements from GPS or THEMIS for example.

However, we can still present an example of results from GREEN-p and compare them to AP8, even if only OPAL-V2 is integrated in the global model. Figure 19 represents protons fluxes versus L* resulting from GREEN-p and AP8 MIN at two magnetic latitudes corresponding to $\alpha_{eq}=90°$ and $\alpha_{eq}=50°$, for E >80 MeV protons. This figure shows that fluxes from

15 GREEN-p come from OPAL-V2 up to L*=1.3 for $\alpha_{eq}=90°$ and up to L*=1.5 for $\alpha_{eq}=50°$ and from AP8 beyond. At very low L*, when AP8 and OPAL-V2 are available, some small differences appear in the flux. At $\alpha_{eq}=50°$ fluxes for GREEN-p are slightly lower than AP8 MIN.

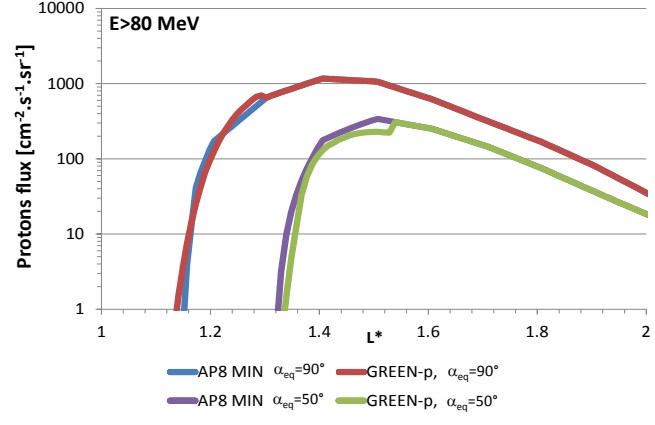



**Figure 19: Protons fluxes versus L\* resulting from GREEN-p and AP8 MIN at two magnetic latitudes corresponding to $\alpha_{eq}=90°$ and $\alpha_{eq}=50°$ for E>80 MeV protons**

## 4 Discussion and conclusions

GREEN (Global Radiation Earth ENvironment) is a new model providing fluxes at any location between L\*=1 and L\*=8 all along the magnetic field lines and for any energy between 1 keV to 10 MeV for electrons and between 1 keV and 800 MeV for protons. This model is composed of global models AE8/AP8, SPM and also local models: SLOT model, OZONE, IGE-2006 for electrons and OPAL and IGP for protons. These local models are used when they are more relevant than AE8/AP8 or SPM. Thus, this version of GREEN is a patch work of existing models with also some improvements, especially at high energy and low L\* values where AE8 model has been corrected, or in the Slot region with the new version of the SLOT model. Obviously, despite our efforts, some discontinuities exist at the interface of the models but will be removed or at least smoothed in the next versions. The major advantage of GREEN is the dependence of fluxes with the solar cycle. Most of models included in GREEN are solar cycle dependent so it allows to have a better estimation of fluxes according to the duration of the mission versus solar cycle. Indeed, fluxes provided by the GREEN model are different for each of the 11 years of the solar cycle. Concerning GREEN-p, which is less finalize than GREEN-e, the major advantage is at low altitude, when OPAL is available, with more than the dependence with the year of the solar cycle but directly a dependence with the radio flux F10.7 of the Sun and the magnetic field of the given year, and also at geostationary orbit with the IGP model. In the next versions of GREEN-p, future studies will allow to predict the magnetic field up to several decades and thus to have a better estimation of the protons fluxes at low altitude. Moreover, in the next future, some efforts will be made to try to extend OPAL model at higher altitude and lower energy by using all the available good quality data (GPS, THEMIS for example), even if we know it would be a hard task.

Another advantage of GREEN is that it is easy to upgrade. Indeed, a cache file system allows switching between models, in order to obtain the most reliable value at each location in space and each energy point. Thus, the way the model is developed is well suited to add new local developments or to include international partnership.

Finally a perspective of GREEN, other than the improvement of flux accuracy would be to develop a special 'worst-case' version of GREEN in order to adapt it to the space industries user needs in the case of short-term missions, typically a few months, such as the case of Electric Orbit Raising missions.

GREEN model would be accessible for space industry in a near future in the OMERE tool (http://www.trad.fr/en/space/omere-sotftware/).

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
