# Peer review of "GREEN: A new Global Radiation Earth Environment model (Beta version)"

_Annales Geophysicae, 2018_

## Referee Comment (RC1) · T.P. OBrien (Referee) · 24 Apr 2018

The GREEN model is a collage of several regional models, mostly those developed by the authors themselves. The models are superimposed upon each other by direct replacement with a priority scheme based on location, energy, and species.

It is described in the manuscript (but not in the title or abstract) as a beta version, meaning it will need additional work before it can be used for satellite design. I concur with publication only minor changes if it is designated as a beta, but I do not think this is made clear in the title or abstract, and it should be made absolutely clear. Satellite designers will be confused or even led astray trying to use this model when it arrives in OMERE if it is not brought up to a higher level of quality or more obviously indicated

as a beta, not yet ready to be used for design.

As a developer of the AE9/AP9-IRENE global radiation belt climatology models, I was very interested to see whether this team of very gifted scientists has developed any new ideas to address the problems that AE9/AP9-IRENE continues to wrestle with. Unfortunately, at this stage of development of GREEN, its synthesis of the different constituent models is very superficial, meaning the underlying challenge of producing a truly global model has not really been attempted. As shown in figures 13 and 14, the GREEN model has large, sharp discontinuities at the boundaries of the underlying models. This will lead to strange results for orbit surveys looking at variations with altitude or inclination. Further, the treatment of temporal variability is similarly superficial: "worst cases" are taken over whatever duration of time the source data provided, and so cannot be applied to the user's mission duration with any corresponding confidence level. How will the engineer know whether to add additional margin (and how much) on top of the model output?

The data-model disagreements shown in Figures 15-18 are comparable to the ones that apparently lead AE9/AP9-IRENE to be "very controversial." How are we to know whether the first non-beta release of GREEN will actually resolve these discrepancies? I suspect from my own experience with AE9/AP9-IRENE that some discrepancies are essentially impossible to resolve definitively because the underlying data sets themselves do not agree. This means a more robust approach to model errors will be required.

I am also uncomfortable with the inner zone correction. It appears to be rather ad hoc. The Boscher et al 2017 paper (which is a very nice paper! and is now available on the IEEE explore website) only really looked at observations of one energy channel. Likewise, the IGP model of protons at geostationary orbit is rather ad hoc and has not been validated. I agree with the authors that these models make "reasonable" assumptions and extrapolations, but they need to be validated somehow. Also, it would be good to cite the RBSP papers by Li (doi:10.1002/2014JA020777) Claudepierre (doi:10.1002/2016JA023719) for the inner zone electrons.

This is a fair start to what will likely be a long and challenging effort to build a truly global model. The paper needs to be a bit more straightforward about the model's state of progress. There is still much work to do. But, this is real progress worthy of publication.

Minor points:

In the abstract, I suggest noting that the model covers all local times (in addition to "all along the magnetic field lines.")

In the introduction, first sentence, add "satellite" before industry.

Page 2, line 2 "no doubt that the obtained averages are more accurate than AP8 and AE8..." I hope the authors are correct! But, they should include AE8/AP8 and probably AE9/AP9-IRENE v1.5 in figures 15-18.

Section 2.1: L* and B/Beq are incompatible coordinates. L* is a drift invariant, but varies with B/Beq on any given field line, while B/Beq varies around the drift orbit. This means that particles with different coordinates are being mixed together depending on where they are measured. Replacing B/Beq with Bmirror solves this problem (or, as Selesnick doi.org/10.1002/2017JA024661 has done, mapping Bmirror and L* to a dipole value of B/Beq. I believe Salammbo does something similar).

What value of F10.7 should be used when running the OPAL model? Is there a long-term forecast built in for using it to design satellites to be flown in the future? This should be discussed, especially since LEO proton fluxes are anti-correlated with F10.7, and some naive users might attempt to be conservative by inputting a very large value of F10.7.

I do not think Figure 18 supports the claim on page 15, line 15 that, "If the period of time of in-situ measurements is long enough (several solar cycles) or is representative of a mean flux, data will easily be compared to GREEN- results." What it tells me, instead,

is that the model-data disagreement can be factor of 4 or more, even when matching the solar cycle phase of the model and data. This suggests either a systematic issue with the model, the data, or something that must be addressed with error bars on one or the other.

---

## Referee Comment (RC2) · Anonymous Referee #2 · 26 Apr 2018

General Comments:

This manuscript reviews the features of many previous radiation belt models including AE8/AP8, SLOT, OZONE, etc, as well as discusses about their advantages and drawbacks. Then they introduce a new comprehensive radiation belt model named GREEN (Global Radiation Earth ENvironment). This model is an integration of the previous models and also can be thought of as a correction to the AE8/AP8 models. GREEN selects the most suitable models for different regions and particle species/energies and combine them, which helps save users' time and effort to select models for their studies. As a user of the ONERA library, I appreciate the authors' effort to build such a complete model and hopefully it will be released with a user friendly interface in the future. I think this work is definitely valuable and interesting, but some revisions need

to be made before publication.

Specific Comments:

The problem that bothers me most is the discontinuity between different regions where the boundary of two sub-models (the models that is used in GREEN, such as AE8 and SLOT) exists. The sharp boundary between AE8 correct and SLOT at about L*=2.5 in Figure 13 is an example. The authors have realized this problem and stated in the paper that this problem is to be solved in the next version. Could you please at least explain how you are going to do it or show some of the potential methods to solve this problem? I suggest to show some preliminary result even if you use very simple smoothing methods such doing running average on each dimensions and repeat it for a few times. Dealing with the boundaries is a critical step in the work of combing different things.

Another thing is that I didn't notice any clear statement in the paper that describes the input and output parameters. I believe for most of the users like me would like to get this kind of information quickly and directly. The first sentence in the abstract tells us the scope of the model but it doesn't say that the solar cycle variation is included too. I suggest to add some more descriptions on the input and output parameters as well as the scope of the model in the abstract.

Here are some more detailed questions and comments:

Page 1/ line 29: asses -> assess

Page 4/ line 7-8: Did you interpolate in between the AE8 MAX and AE MIN and if so, how did you do it?

Figure 3: In the paragraph above this figure, it is mentioned that AE8 corrected is applied for >1 MeV electrons. But in this figure, it looks that the correction starts at about 700 keV. Could you please clarify that in the text?

Page 5/ line 3: It is not clear just to say "a mean model". I suggest to change it to

something like "a model that reflects the mean flux along the magnetic field lines".

Figure 5, 6 and many other figures: There are too many lines in the figure which makes it hard for the readers to see the data clearly. It would be nice if you could reduce the lines you plot on one figure. Figure 5 has a large blank at the top and I suggest to change the y axis range so that the data is shown more clearly. Also, L star is used in the paper, but in some of the figures it is shown as "L". It would be better if they are accurate and shown as "L*". For figure 6, there is no y axis for the F10.7 index and it is better if there is one on the right or just plot it as a separate panel.

Page 7/ line 5: provide->provides; maximum flux-> fluxes

Page 8/ line 19: How is this extrapolation done? How did you choose the fluxes at the lower energy boundary.

Page 9/ line 8: used also -> also used

Page 10/ line 10: which lies "below" the general tendency of the curve.

Figure 10, could you please shrink the y axis range to make the variation of the lines clearer. Same thing with Figure 11.

Page 12/ line 13: non trapped -> untrapped (to be consistent with the one on Page 11)

Figure 16: Since NOAA satellite is at low altitude, it will only measure the particles with really small pitch angles at large L shells. But GREEN provides an average flux of all pitch angles. Most of the times, the pitch angle distribution peaks at 90 degree, so I would expect NOAA lines generally lie below the GREEN lines. But this is not the case especially for the >30 keV electrons. Could you please comment on this?

Page 17/ line 18: protons-> proton fluxes

Page 17/ line 19: extend OPAL model at-> to higher altitude

---

## Author Comment (AC1) · 7 Jun 2018

Modifications compared to the first version of the paper are highlighted in red in the text.

Referee Number 1:

General Comments: This manuscript reviews the features of many previous radiation belt models including AE8/AP8, SLOT, OZONE, etc, as well as discusses about their advantages and draw-backs. Then they introduce a new comprehensive radiation belt model named GREEN (Global Radiation Earth ENvironment). This model is an integration of the previous models and also can be thought of as a correction to the AE8/AP8 models. GREEN selects the most suitable models for different regions and

particle species/energies and combine them, which helps save users' time and effort to select models for their studies. As a user of the ONERA library, I appreciate the authors' effort to build such a complete model and hopefully it will be released with a user friendly interface in the future. I think this work is definitely valuable and interesting, but some revisions need to be made before publication.

Specific Comments: The problem that bothers me most is the discontinuity between different regions where the boundary of two sub-models (the models that is used in GREEN, such as AE8 and SLOT) exists. The sharp boundary between AE8 correct and SLOT at about L*=2.5 in Figure 13 is an example. The authors have realized this problem and stated in the paper that this problem is to be solved in the next version. Could you please at least explain how you are going to do it or show some of the potential methods to solve this problem? I suggest to show some preliminary result even if you use very simple smoothing methods such doing running average on each dimensions and repeat it for a few times. Dealing with the boundaries is a critical step in the work of combing different things.

==>Yes, you're right, there are some discontinuities at the interface of the different models integrated in GREEN and this is a critical step in this work. In the paper I have added an example of results with a simple smoothing method (Figure 14 in the new version) and some comments about the discontinuities.

Another thing is that I didn't notice any clear statement in the paper that describes the input and output parameters. I believe for most of the users like me would like to get this kind of information quickly and directly. The first sentence in the abstract tells us the scope of the model but it doesn't say that the solar cycle variation is included too. I suggest to add some more descriptions on the input and output parameters as well as the scope of the model in the abstract.

==>The inputs and outputs are clearly described in Figure 1 and the solar cycle dependence is mentioned. However I add some more descriptions in the abstract.

Here are some more detailed questions and comments: Page 1/ line 29: asses -> assess

==>Ok, corrected

Page 4/ line 7-8: Did you interpolate in between the AE8 MAX and AE8 MIN and if so, how did you do it?

==>No interpolation is done between the AE8 MAX and AE8 MIN. On one solar cycle (11 years) with 0 the year of the minimum, if -2 ïĆč year <2, AE8 MIN is used, otherwise AE8 MAX is used.

Figure 3: In the paragraph above this figure, it is mentioned that AE8 corrected is applied for >1 MeV electrons. But in this figure, it looks that the correction starts at about 700 keV. Could you please clarify that in the text?

==>The correction of AE8 is for high energy electron, typically greater than about 1 MeV. But, actually this correction is applied for electron greater then few hundred of keV, depending on Lvalue. So I have modified it in the paragraph above Figure 3.

Page 5/ line 3: It is not clear just to say "a mean model". I suggest to change it to something like "a model that reflects the mean flux along the magnetic field lines".

==>I have modified the sentence in the text by "that reflects the mean flux at each point along the magnetic field lines".

Figure 5, 6 and many other figures: There are too many lines in the figure which makes it hard for the readers to see the data clearly. It would be nice if you could reduce the lines you plot on one figure. Figure 5 has a large blank at the top and I suggest to change the y axis range so that the data is shown more clearly. Also, L star is used in the paper, but in some of the figures it is shown as "L". It would be better if they are accurate and shown as "L*". For figure 6, there is no y axis for the F10.7 index and it is better if there is one on the right or just plot it as a separate panel.

==>Figure 5 and Figure 5 have been modified for a better readability.

Page 7/ line 5: provide->provides; maximum flux-> fluxes

==>ok

Page 8/ line 19: How is this extrapolation done? How did you choose the fluxes at the lower energy boundary.

==>After reflection, this extrapolation is not of great interest and does not bring anything. Moreover, this extrapolation should not be used in a specification model.

Page 9/ line 8: used also -> also used

==>ok

Page 10/ line 10: which lies "below" the general tendency of the curve.

==>ok

Figure 10, could you please shrink the y axis range to make the variation of the lines clearer. Same thing with Figure 11.

==>Figure 10 and Figure 11 have been re-plotted.

Page 12/ line 13: non trapped -> untrapped (to be consistent with the one on Page 11)

==>ok

Figure 16: Since NOAA satellite is at low altitude, it will only measure the particles with really small pitch angles at large L shells. But GREEN provides an average flux of all pitch angles. Most of the times, the pitch angle distribution peaks at 90 degree, so I would expect NOAA lines generally lie below the GREEN lines. But this is not the case especially for the >30 keV electrons. Could you please comment on this? I think there are some misunderstandings in Figure 16. First, you're right, NOAA spacecraft measure particles with small pitch angles at large L-shell. But, you misunderstood when you say that GREEN provides an average flux of all pitch angles. It is not the

case! GREEN provides flux for each pitch angle along the magnetic field line. It is not an average of all pitch angles. Fluxes provided by GREEN are higher at magnetic equator than for small pitch angles. Figure 16 plots electrons flux provided by GREEN at LEO orbit, so they can be directly compared to NOAA measurements. If GREEN model and NOAA data were perfect, fluxes from GREEN would be superposed with NOAA measurements on this plot.

==>Then, it is important to keep in mind than >30 keV electron fluxes come from AE8 in GREEN while higher energy fluxes come from Slot or OZONE models. So the big difference between NOAA measurements and GREEN fluxes for >30 keV electron come from AE8 model, which seems to underestimate fluxes for L*>4.

Page 17/ line 18: protons-> proton fluxes

==>ok

Page 17/ line 19: extend OPAL model at-> to higher altitude

==>ok

Please also note the supplement to this comment:
https://www.ann-geophys-discuss.net/angeo-2018-26/angeo-2018-26-AC1-supplement.pdf
* * *
[Figure]

**Supplement:**

[revised manuscript text omitted]

---

## Author Comment (AC2) · 7 Jun 2018

The GREEN model is a collage of several regional models, mostly those developed by the authors themselves. The models are superimposed upon each other by direct replacement with a priority scheme based on location, energy, and species. It is described in the manuscript (but not in the title or abstract) as a beta version, meaning it will need additional work before it can be used for satellite design. I concur with publication only minor changes if it is designated as a beta, but I do not think this is made clear in the title or abstract, and it should be made absolutely clear. Satellite designers will be confused or even led astray trying to use this model when it arrives in OMERE if it is not brought up to a higher level of quality or more obviously indicated as a beta, not yet ready to be used for design.

[Figure]

==>The term "beta version" has been added in the title of the paper.

As a developer of the AE9/AP9-IRENE global radiation belt climatology models, I was very interested to see whether this team of very gifted scientists has developed any new ideas to address the problems that AE9/AP9-IRENE continues to wrestle with. Unfortunately, at this stage of development of GREEN, its synthesis of the different constituent models is very superficial, meaning the underlying challenge of producing a truly global model has not really been attempted. As shown in figures 13 and 14, the GREEN model has large, sharp discontinuities at the boundaries of the underlying models. This will lead to strange results for orbit surveys looking at variations with altitude or inclination.

==>Yes, you're right, there are some sharp discontinuities at the boundaries of the underlying models. A comment of the other reviewer was also on these discontinuities so I add in the paper some comments about this point. I try to show what can be done with a very simple smooth function but obviously it is not so easy. For now, we do not know how to attenuate discontinuities otherwise than improving each of the underlying models. As you well know, it is a critical step in the development of a global model.

Further, the treatment of temporal variability is similarly superficial: "worst cases" are taken over whatever duration of time the source data provided, and so cannot be applied to the user's mission duration with any corresponding confidence level. How will the engineer know whether to add additional margin (and how much) on top of the model output?

==>It is a choice on our part to not take into account confidence level in GREEN model. For now, we try to answer as best as possible to the first need of space industries, which is a "mean" model. We do not pretend to make a "worst case" model for internal charging for example. The notion of "maximum mean flux" in the outputs of GREEN is just an upper envelop which take into the variations of flux from one solar cycle to another (as in the case of IGE-2006). The notion of worst-case is not addressed in his

paper.

The data-model disagreements shown in Figures 15-18 are comparable to the ones that apparently lead AE9/AP9-IRENE to be "very controversial." How are we to know whether the first non-beta release of GREEN will actually resolve these discrepancies? I suspect from my own experience with AE9/AP9-IRENE that some discrepancies are essentially impossible to resolve definitively because the underlying data sets themselves do not agree. This means a more robust approach to model errors will be required.

==>You make a point...GREEN is probably as "controversial" as AE9/AP9-IRENE at the boundaries of the underlying models. I remove form my paper the word "controversial" that seems to bother you. You're right, some discrepancies are very difficult to resolve. We hope in a future to smooth GREEN model using our physical model Salammbô. But the way we could do that needs to be defined.

I am also uncomfortable with the inner zone correction. It appears to be rather adhoc. The Boscher et al 2017 paper (which is a very nice paper! and is now available on the IEEE explore website) only really looked at observations of one energy channel. Likewise, the IGP model of protons at geostationary orbit is rather ad hoc and has not been validated. I agree with the authors that these models make "reasonable" assumptions and extrapolations, but they need to be validated somehow. Also, it would be good to cite the RBSP papers by Li (doi:10.1002/2014JA020777) Claude-Pierre (doi:10.1002/2016JA023719) for the inner zone electrons.

==>For the inner zone electrons, I add the two references you mentioned. For the IGP model, it is really difficult to validate it because there are no good proton data at GEO. CRRES/MEB and THEMIS/SST are contaminated, RBSP has no data at GEO. So unfortunately, I can not validate IGP model.

This is a fair start to what will likely be a long and challenging effort to build a truly global model. The paper needs to be a bit more straightforward about the model's

state of progress. There is still much work to do. But, this is real progress worthy of publication.

Minor points: In the abstract, I suggest noting that the model covers all local times (in addition to "all along the magnetic field lines.").

==>Ok

In the introduction, first sentence, add "satellite" before industry.

==>Ok

Page 2, line 2 "no doubt that the obtained averages are more accurate than AP8 and AE8..." I hope the authors are correct! But, they should include AE8/AP8 and probably AE9/AP9-IRENE v1.5 in figures 15-18.

==>Comparison with AE9 in Figure 16 and Figure 17 and some comments have been added in the paper.

Section 2.1: L* and B/Beq are incompatible coordinates. L* is a drift invariant, but varies with B/Beq on any given field line, while B/Beq varies around the drift orbit. This means that particles with different coordinates are being mixed together depending on where they are measured. Replacing B/Beq with Bmirror solves this problem (or, as Selesnick doi.org/10.1002/2017JA024661 has done, mapping Bmirror and L* to a dipole value of B/Beq. I believe Salammbo does something similar).

==>I am not sure to understand what you want to say. Actually, in the GREEN model, we use the 3D grid of Salammbô, in Ec, yeq=sin($\alpha$eq) and L*. But, as the term yeq or even the term "pitch angle" is not always known or understood by the reader, we have chosen to speak of B/Beq in the paper. When we develop a model using in-situ data, we consider that a given location (x,y,z) corresponds to a couple (L*, $\alpha$eq). $\alpha$eq is calculated by considering the first invariant such as $\sin^2(\alpha$eq)/Beq = $\sin^2(\alpha$mirror)/Bmirror in which we consider that Blocal at x,y,z is Bmirror.

What value of F10.7 should be used when running the OPAL model? Is there a long-term forecast built in for using it to design satellites to be flown in the future? This should be discussed, especially since LEO proton fluxes are anti-correlated with F10.7, and some naive users might attempt to be conservative by inputting a very large value of F10.7.

==>These comments have been added in the paper: "As OPAL depends on the radio flux F10.7, an input of OPAL is the date. So, for a given date chosen by the user in the past, the real F10.7 value is used to calculate proton fluxes. But for a given date in the future, it is not so easy because the F10.7 value is unknown. Consequently, a statistical study has been done on F10.7 values from 1947 to now in order to define a mean F10.7 value for each of the eleven years of a solar cycle. Thus, for a given date chosen by the user in the future, the year of the solar cycle is predicted (from year -6 to year +4, 0 being the year of the minimum) and according to this one, the corresponding mean F10.7 value is used in OPAL to calculate proton fluxes. Moreover, added to the mean protons fluxes, OPAL provides an upper envelop considering the variation from one solar cycle to another. Taking into account that high energy proton fluxes are anti-correlated with F10.7 values, this upper envelop is calculated using the minimum of F10.7 value measured since 1947 for each year of solar cycle."

I do not think Figure 18 supports the claim on page 15, line 15 that, "If the period of time of in-situ measurements is long enough (several solar cycles) or is representative of a mean flux, data will easily be compared to GREEN- results." What it tells me, instead, is that the model-data disagreement can be factor of 4 or more, even when matching the solar cycle phase of the model and data. This suggests either a systematic issue with the model, the data, or something that must be addressed with error bars on one or the other.

==>What I wanted to say here is that differences between mean model and data can not be due to a problem in the model or in the data but just in the fact that the time period of the data can correspond to a calm period in term of magnetic activity or flux. So

using short term data (few years only) to develop a model can be dangerous because the time period of the data (active or calm) can have significant consequences on the resulting model and can not be representative of a mean flux. However, as a model developer, I am fully aware of the difficulty of having good quality data for a long period of time. However I agree with you when you said that "the model-data disagreement can be factor of 4 or more, even when matching the solar cycle phase of the model and data". Actually, due to the variation of flux from one solar cycle to another, even when the solar cycle phase is the same some differences can be observed between data and model. This is why, as it has been done in IGE-2006 model, we try to estimate an upper envelop of flux taking into account the variation from one solar cycle to another.

---

## Author Response (AR1)

**Response to Editor**

*Thank you for a detailed response to the comments by the two reviewers. Although commenting was extensive, I believed most of the issues raised were minor and I find almost all answers satisfactory. After reviewing your answers and revisions (in the "tracked changes" version of your submitted, revised article), I am happy to recommend your manuscript for publication in Annales Geophysicae, following few technical and minor corrections.:*

*Figure 4: Please generate a higher resolution plot. Parts of the plot (axes, legend, data points) are too fuzzy or blurred. Please also change the title to L\*: 3.7-3.8 (instead of L3.7_3.8). The meaning of first line of the legend (in red) is also not clear.*

*Figure 16: The vertical axis looks cropped, please fix.*

*General on Figures: in most cases, superscript fonts are used to denote the cm-2 s-1 units (for the power value), except in some cases (Figures 4 and 15-17). Please use superscript also in these cases.*

→ I think I have corrected all little imperfections in figures as you mentioned.

*General on units: In some cases the flux is given per steradian, in others integrated over the solid angle. Please verify that in all cases units are consistent – it is not uncommon to have spectra differing by factors of "pi" because an angular/pitch angle integration was forgotten, so I just want to make sure there is such issue. Also, do values given "per steradian" correspond to specific pitch angles or to pitch angle averages of fluxes (taking into account the PAD shape at each energy/L\*)?*

→ I have checked all the figures and the units are ok. In most of the cases "$sr^{-1}$" just comes from omnidirectional data divided by $4\pi$.

*Regarding your answer to reviewer 2 on "Figure 16: Since NOAA satellite is at low altitude, it will only measure the particles with really small pitch angles at large L shells. But GREEN provides an average flux of all pitch angles. Most of the times, the pitch angle distribution peaks at 90 degree, so I would expect NOAA lines generally lie below the GREEN lines. But this is not the case especially for the >30 keV electrons. Could you please comment on this?":*

*Your clarification, that GREEN provides a flux at each pitch angle, is useful. However, I suspect that what reviewer was maybe inquiring is how do you quantify in GREEN the flux at large pitch angles and L-shells, given that at large L-shells NOAA cannot monitor large equatorial pitch angles. Do you assume a certain PAD shape which you constrain with NOAA observations at low pitch angles?*

→ Yes we assume a PAD shape. I have specified it in the text: "An equatorial pitch angle distribution shape in sinus is assumed and constrained by data all along the magnetic field lines between L\*=2.5 and L\*=5 (Figure 5)."

*On the issue of discontinuities (p. 13, lines 15-19 in the revised article): You suggest that improving each model at each boundaries may be a solution, but wouldn't that be unnecessary complex (ie. to improve 6 models in parallel?). Why can there be no consistent way develop a unified model across many L\* and energies, combining in parallel all different measurements that each of the 6 "submodels" of GREEN. I believe you partly address this is the introduction, but maybe you can be more explicit why this has not worked (or cannot work).*

➔ I have added a sentence in the introduction to be more clear: "Obviously, the ideal would be to develop a unified global model across many L* and energies rather than combining "submodels". However, radiation belts are made of several regions with different dynamics and several populations (low energies and high energies) with different behavior. So it is easier to develop local models for each region and each energy range.

*Non-public comments to the author:*

*Dear Dr. Sicard,*

*thank you for submitting your manuscript to Annales Geophysicae. After reviewing the article myself and your discussion with the referees, I am glad to proceed with its acceptance for publication, after some technical and minor issues are clarified, as you can see in my public comments.*

*Kind regards,*

*Elias Roussos*

➔ Thank you Elias for taking the time to read and reread this paper.

Kind regards,

Angélica